# Label Consistency-based Worker Filtering for Crowdsourcing

**Jiao Li**[1]          **Liangxiao Jiang**[*1]          **Chaoqun Li**[2]          **Wenjun Zhang**[1]

[1]School of Computer Science, China University of Geosciences, Wuhan 430074, China.
[2]School of Mathematics and Physics, China University of Geosciences, Wuhan 430074, China.

## Abstract

In crowdsourcing scenarios, we can obtain multiple noisy labels from different crowd workers on the Internet for each instance and then infer its unknown true label via a label integration method. However, noisy labels often have a serious negative impact on label integration. In this case, most existing works always focus on designing more complex label integration methods to infer unknown true labels more accurately from multiple noisy labels, but little attention has been paid to another perspective, i.e., purifying noisy labels before label integration. In this paper, we aim to purify noisy labels for existing label integration methods and propose a label consistency-based worker filtering (LCWF) algorithm. In LCWF, we consider that if all low-quality workers are filtered out and only high-quality workers remain, the label consistency should be high. Therefore, we utilize label consistency to filter out low-quality workers. Firstly, we directly transform the worker filtering problem into a discrete optimization problem and utilize label consistency to define the fitness function for this problem. Then, we search for the optimal solution to this problem by a genetic algorithm. Finally, we filter out all labels from low-quality workers according to the optimal solution we obtained. Experimental results on simulated and real-world datasets demonstrate that LCWF can effectively purify noisy labels and improve the integration accuracy of existing label integration methods.

## 1   INTRODUCTION

Supervised learning has been proven to be incredibly powerful in various fields due to its ability to learn from labeled

---

*Corresponding author, Liangxiao Jiang <ljiang@cug.edu.cn>

data [Jiang et al., 2019a, Zhang et al., 2023a]. In supervised learning, a large amount of high-quality labeled data is an essential requisite for effective model learning [Wang and Wu, 2021, Chen et al., 2022b]. Traditionally, labeling tasks are primarily performed by experts in the field or well-trained workers. The labels obtained by this way show high quality, but result in high economic and time costs [Li et al., 2016, Xu et al., 2021, Zhu et al., 2023].

Fortunately, the advent of crowdsourcing has provided a cost-effective and convenient way for extensive label collection [Liang et al., 2022, Zhang, 2022]. Through crowdsourcing platforms [Buhrmester et al., 2011] such as Crowdflower, Click-worker and Amazon Mechanical Turk (AMT), we can post crowdsourcing tasks and then obtain a large number of labels from hired crowd workers at a low cost [Chen et al., 2022a, Tong et al., 2020]. However, due to the lack of experience, well-training, etc., the quality of labels obtained from an individual crowd worker on the Internet is usually poor [Rodrigues and Pereira, 2018, Chen et al., 2020]. To address this problem, repeated labeling is proposed [Sheng et al., 2008], in which multiple noisy labels are provided by different crowd workers for each instance and then its unknown true label can be inferred via a label integration method [Ma et al., 2015].

In the past few years, although many works have proved the effectiveness of label integration, the integration accuracy of existing label integration methods is still far from perfect. This is primarily due to the serious negative impact from noisy labels. In this case, lots of efforts have been made by researchers on designing more complex label integration methods to infer unknown true labels more accurately. These label integration methods include: majority voting (MV) [Sheng et al., 2008], Dawid-Skene (DS) [Dawid and Skene, 1979], ZenCrowd (ZC) [Demartini et al., 2012], generative model of labels, abilities, and difficulties (GLAD) Whitehill et al. [2009], ground truth inference using clustering (GTIC) [Zhang et al., 2016], iterative weighted majority voting (IWMV) [Li and Yu, 2014], label similarity-based weighted soft majority voting (LSWSMV) [Tao et al., 2020],

differential evolution-based weighted soft majority voting (DEWSMV) [Tao et al., 2021], multiple noisy label distribution propagation (MNLDP) [Jiang et al., 2022], label augmented and weighted majority voting (LAWMV) [Chen et al., 2022c], attribute augmentation-based label integration (AALI) [Zhang et al., 2023b], instance redistribution-based label integration (IRLI) [Zhang et al., 2024], etc.

As mentioned above, most existing works always focus on designing more complex label integration methods to infer unknown true labels more accurately from multiple noisy labels, but little attention has been paid to another perspective, i.e., purifying noisy labels before label integration. In theory, noisy labels are usually from low-quality workers. If we are able to filter out all low-quality workers, we can purify most of the noisy labels and subsequently reduce their negative impact on label integration. Therefore, in this paper, we aim to purify noisy labels by worker filtering to improve the performance of existing label integration methods.

To this end, we propose a label consistency-based worker filtering (LCWF) algorithm. In LCWF, we consider that if all low-quality workers are filtered out and only high-quality workers remain, the label consistency should be high. Therefore, we utilize label consistency to filter out low-quality workers. Firstly, we directly transform the worker filtering problem into a discrete optimization problem and utilize label consistency to define the fitness function for this problem. Then, we search for the optimal solution to this problem by a genetic algorithm. Finally, we filter out all labels from low-quality workers according to the optimal solution we obtained. In general, the contributions of this work can be summarized as follows:

1. We propose to purify noisy labels before label integration to infer unknown true labels more accurately. Different from most existing works focusing on designing more complex label integration methods with better performance, we aim to purify noisy labels for existing label integration methods, which provides a new perspective for improving the performance of label integration.

2. We propose a label consistency-based worker filtering (LCWF) algorithm to purify noisy labels for existing label integration methods. In LCWF, we transform the worker filtering problem into a discrete optimization problem and utilize label consistency to define the fitness function. Then, a genetic algorithm is used to search for the optimal solution to this problem.

3. We conduct extensive experiments to evaluate the proposed LCWF on real-world and simulated datasets. The experimental results show that LCWF can effectively purify noisy labels and improve the integration accuracy of existing label integration methods.

The rest of this paper is organized as follows. Section 2 describes the work related to this paper. Section 3 introduces the proposed LCWF in detail. Section 4 reports the experiments and results on real-world and simulated datasets. Section 5 summarizes this paper and outlines the research directions of future work.

## 2 RELATED WORK

To infer unknown true labels from multiple noisy labels in crowdsourcing, label integration has attracted a great deal of attention from researchers. A variety of label integration methods have been proposed for better performance [Sheng and Zhang, 2019].

MV [Sheng et al., 2008] is the simplest label integration method, which regards the label with the highest votes as the integrated label directly. However, MV ignores the difference in the label quality of crowd workers, so its performance is very limited. After MV, more complex label integration methods have been proposed one after another for better performance. Specifically, DS [Dawid and Skene, 1979] jointly estimates the confusion matrix of each worker and the integrated label of each instance by the EM algorithm [Singh, 2006]. ZC [Demartini et al., 2012] uses a two-element parameter to iteratively estimate the reliability of each worker. GLAD [Whitehill et al., 2009] builds a probability model to infer the integrated labels, the labeling difficulty of instances and the professional level of crowd workers. GTIC [Zhang et al., 2016] uses the K-means algorithm to cluster all instances into distinct clusters and then assigns the same class label to instances within the same cluster. IWMV [Li and Yu, 2014] first uses original integrated labels obtained by MV to estimate the label quality of each worker, and then iteratively updates both the label quality and the integrated label.

In recent years, more label integration methods have been proposed from various novel perspectives, and they have further improved the performance of label integration. For example, DEWSMV [Tao et al., 2021] defines three objective functions to optimize the label quality of workers when labeling different instances by a differential evolution (DE) algorithm. Jiang et al. [2022] propose MNLDP, which first converts multiple noisy labels to multiple noisy label distributions and then optimizes the weights of the nearest neighbors. Finally, each instance obtains a portion of multiple noisy label distributions from its nearest neighbors while retaining a portion of its own. Chen et al. [2022c] propose LAWMV, which finds neighbors by the KNN algorithm to augment each instance's multiple noisy labels and obtains the integrated labels by weighted majority voting. Zhang et al. [2023b] propose AALI, which designs an attribute augmentation method to enrich the original attribute space and builds multiple component classifiers on reliable instances to predict the integrated labels.

Although label integration is often effective, a certain level

of noise still remains in the integrated labels. To our knowledge, since Li et al. [2016] propose to employ noise filters to filter the noise in integrated labels, plentiful works has been developed on noise handling after label integration. Noise handling consists of two steps: noise filtering and noise correction. Representative works on noise filtering include: CF [Gamberger et al., 1999], MVF [Brodley and Friedl, 1999], IPF [Khoshgoftaar and Rebours, 2007], etc. Representative works on noise correction include: PL [Nicholson et al., 2016], STC [Nicholson et al., 2016], CTNC [Dong et al., 2022], DVNC [Ji et al., 2023], etc.

However, the goal of these works is to purify the noise that is still present in the integrated labels after label integration, and the work aiming to purify the noise in the crowd labels before label integration is still limited. As we have analyzed, filtering low-quality workers can be an effective way to purify noisy labels. There exists a few works focus on worker filtering in crowdsourcing. For example, Raykar and Yu [2012] define the worker who assigns labels randomly for each instance as a spammer, and then propose an empirical Bayesian algorithm called SpEM to iteratively filter out the spammers. However, low-quality workers are not just likely to provide random labels. Therefore, this method cannot identify all low-quality workers. In addition, Dekel and Shamir [2009] evaluate the quality of each worker and filter out the low-quality ones with a very simple algorithm. But they only focus on the setting of a single noisy label per instance and binary classification problems. Therefore, in this paper, we aim to purify noisy labels for existing label integration methods and propose a label consistency-based worker filtering (LCWF) algorithm. Different from above works on worker filtering, LCWF considers low-quality workers without specific assumptions on their labeling strategy and is applicable to multi-class classification problems. We will describe LCWF in detail in Section 3.

# 3 LABEL CONSISTENCY-BASED WORKER FILTERING

## 3.1 MOTIVATION

In crowdsourcing scenarios, a crowdsourced dataset is usually denoted by a set $D = \{(x_i, L_i)\}_{i=1}^{N}$, where $N$ denotes the number of all instances in $D$, $x_i$ denotes the $i$-th instance and $L_i = \{l_{ir}\}_{r=1}^{R}$ denotes a multiple noisy label set associated with $x_i$. In $L_i$, $l_{ir}$ denotes the label of $x_i$ from the worker $u_r$ $(r = 1, 2, \ldots, R)$ and takes the value from a fixed set $\{c_1, c_2, \ldots, c_Q, -1\}$, where $R$ and $Q$ denote the number of all workers and all classes, respectively. When $l_{ir}$ takes as -1, it denotes that $u_r$ does not label $x_i$. To infer the unknown true label $y_i$ for the instance $x_i$, label integration is usually used to obtain an integrated label $\hat{y}_i$ from $L_i$, which is expected to be as consistent as possible with $y_i$.

Table 1: Multiple noisy labels of five instances.

|       | $u_1$ | $u_2$ | $u_3$ | $u_4$ | $u_5$ |
|-------|-------|-------|-------|-------|-------|
| $x_1$ | $c_1$ | $c_1$ | $c_1$ | $c_2$ | $c_1$ |
| $x_2$ | $c_1$ | $c_1$ | $c_2$ | $c_3$ | $c_2$ |
| $x_3$ | $c_1$ | $c_1$ | $c_1$ | $c_1$ | $c_3$ |
| $x_4$ | $c_1$ | $c_2$ | $c_1$ | $c_1$ | $c_3$ |
| $x_5$ | $c_1$ | $c_1$ | $c_1$ | $c_1$ | $c_2$ |

Table 2: Multiple noisy labels after filtering out a low-quality worker $u_5$.

|       | $u_1$ | $u_2$ | $u_3$ | $u_4$ |
|-------|-------|-------|-------|-------|
| $x_1$ | $c_1$ | $c_1$ | $c_1$ | $c_2$ |
| $x_2$ | $c_1$ | $c_1$ | $c_2$ | $c_3$ |
| $x_3$ | $c_1$ | $c_1$ | $c_1$ | $c_1$ |
| $x_4$ | $c_1$ | $c_2$ | $c_1$ | $c_1$ |
| $x_5$ | $c_1$ | $c_1$ | $c_1$ | $c_1$ |

Table 3: Multiple noisy labels after filtering out a high-quality worker $u_1$.

|       | $u_2$ | $u_3$ | $u_4$ | $u_5$ |
|-------|-------|-------|-------|-------|
| $x_1$ | $c_1$ | $c_1$ | $c_2$ | $c_1$ |
| $x_2$ | $c_1$ | $c_2$ | $c_3$ | $c_2$ |
| $x_3$ | $c_1$ | $c_1$ | $c_1$ | $c_3$ |
| $x_4$ | $c_2$ | $c_1$ | $c_1$ | $c_3$ |
| $x_5$ | $c_1$ | $c_1$ | $c_1$ | $c_2$ |

As mentioned above, in this paper, we aim to purify noisy labels before label integration to improve the performance of existing label integration methods. As noisy labels usually from low-quality workers, we consider to purify noisy labels by filtering out low-quality workers. So the key problem we have to solve is how to identify the low-quality workers.

For this purpose, we first anticipate the status when only high-quality workers remain and then attempt to achieve this status. Specifically, since it is easier to reach a consensus among high-quality workers, it is natural to note that if all low quality workers are filtered out and only high-quality workers remain, the label consistency should be high. In other words, the multiple noisy labels of each instance should be highly consistent. Let us illustrate this with a concrete example. Assume that there exists five instances $\{x_i\}_{i=1}^{5}$ from a multi-class dataset, and the true label of these instances is $c_1$. Five different workers $\{u_r\}_{r=1}^{5}$ have labeled them, which are shown in Table 1, where we use red to mark the correct labels and blue to mark the incorrect labels. It is obvious that the worker $u_5$ is more likely to be a low-quality worker since most of the labels from $u_5$ are incorrect. And other workers are more likely to be high-quality workers since most of their labels are correct. If the low-quality worker $u_5$ is filtered out, the multiple noisy labels become more consistent, as shown in Table 2. On the contrary, if a high-quality worker $u_1$ is filtered out, the multiple noisy labels become less consistent, as shown in

Table 3. Therefore, we can conclude that if higher label consistency is obtained after worker filtering, the workers who are filtered out are more likely to be low-quality workers.

It should be noted that, this conclusion is based on the assumption that there are more correct labels than incorrect labels. In fact, for a valuable crowdsourcing dataset, the proportion of correct labels really should be larger than the proportion of incorrect labels, and only based on this MV and most label integration methods can be performed effectively [Sheng et al., 2008, Karger et al., 2014]. So in this general crowdsourcing scenarios we focus on, the conclusion above can be made. Inspired by this conclusion, we attempt to identify low-quality workers by maximizing the label consistency after worker filtering. At this point, we have transformed the worker filtering problem into a discrete optimization problem with the goal of maximizing the label consistency after worker filtering.

In the past years, how to solve discrete optimization problems has already been widely studied, and various corresponding algorithms have been proposed, among which genetic algorithm (GA) [Holland, 1992] is a well-known one. Due to its powerful characteristics, GA has been applied successfully to a variety of discrete optimization problems, and various variants of GA have been proposed by researchers, which can broadly be classified into five main categories: real and binary coded [Shukla et al., 2019], multi-objective [Emmerich and Deutz, 2018], parallel [Harada and Alba, 2021], chaotic [Wang and Sobey, 2020], and hybrid GAs [El-Mihoub et al., 2006]. Although these variants of GA can deal with different scenarios well, them are relatively complex to be applied to solve the problem of this paper. In this paper, we just carry on the standard GA to identify and then filter out low-quality workers.

## 3.2 THE PROPOSED LCWF

How to perform GA to identify the low-quality workers is the core problem we need to solve in this section. Generally speaking, although the details of different implementations of GA is various, they commonly share the same structure as follows: The algorithm operates by iteratively updating a pool of hypotheses, called the population. On each iteration, each member within the population is evaluated according to a fitness function and improved by three evolutionary operators: selection, crossover, and mutation. We follow the general structure of GA and adapt it to identify low-quality workers as follows:

**Initialization:** Firstly, we initialize a pool of hypotheses to form an initial population. We represent a hypothesis by a binary bit string, which corresponds to a solution for worker filtering. Specifically, for the $k$-th hypothesis $\boldsymbol{h}_k$ in the population $P = \{\boldsymbol{h}_k\}_{k=1}^K$, where $K$ denotes the population size, we represent $\boldsymbol{h}_k$ as $\{h_{k1}, h_{k2}, \ldots, h_{kr}, \ldots, h_{kR}\}$,

where $h_{kr} \in \{0, 1\}$. If $h_{kr}$ is 0, it means that the $r$-th worker $u_r$ is identified as a low-quality worker. We randomly generate $K$ hypotheses to $P$. In this way, we obtain the initial population $P$ for the following evolutionary operators.

**Selection:** The selection operator is often used to select a certain proportion of hypotheses from the current population to the next generation or to produce offspring as parents. The probability of a hypothesis being selected is closely related to its fitness. Specifically, the hypothesis with higher fitness is more likely to be selected. In LCWF, we apply the roulette wheel selection as the selection operator. The probability of the hypothesis $\boldsymbol{h}_k$ being selected is calculated as follows:

$$\Pr(\boldsymbol{h}_k) = f(\boldsymbol{h}_k) / \sum_{k=1}^K f(\boldsymbol{h}_k), \tag{1}$$

where $f$ is a fitness function that assigns a fitness for a given hypothesis. According to the probability calculated by Eq.(1), in each generation, we select $(1 - s) * K$ members of $P$ to the next generation directly, where $s$ is the fraction of the population to be replaced by crossover.

**Crossover:** Crossover is a crucial operator for the production of offspring in GA. In the crossover operator, the bit at each position $r$ in each offspring is from the bit at the position $r$ in one of the two parents. In LCWF, for each pair of parents, we produce two offspring by applying the uniform crossover operator. Firstly, for each position $r$, we randomly decide whether we need to swap the bit at the position $r$ in parents. Then, after all bits have been determined, we copy the parents directly to the offspring. According to the probability calculated by Eq. (1), in each generation, we select $s * K/2$ pairs of hypotheses from the population $P$ to produce offspring by the crossover operator, and then we add these offspring to the next generation.

**Mutation:** The mutation operator is a useful approach to maintain the diversity of the population and escape from the local optimum. In each generation, we randomly select $m$ percent of hpotheses to mutate. Specifically, we invert one randomly selected bit in their representation. That is, if the randomly selected bit is 1, we invert it to 0, Otherwise 1.

Now, the only left problem is how to design the fitness function, which is critical to LCWF. In the previous subsection, we have analyzed that the goal of GA is to maximize the label consistency after worker filtering. Thus, our fitness function should satisfy the following principle: For the hypothesis $\boldsymbol{h}_k$ and its corresponding dataset $D_k$, which is constructed by filtering out some workers according to $\boldsymbol{h}_k$, if $\boldsymbol{h}_k$ can lead to high label consistency of $D_k$, it should be given a high fitness, otherwise a low fitness. To satisfy this principle, for the $r$-th remained worker $u_r$ in the dataset $D_k$, we calculate $LC_r$ to measure the label consistency of $u_r$ with other remained workers as follows:

$$LC_r = \frac{1}{|D_{kr}|} \sum_{i=1}^{|D_{kr}|} \sum_{\substack{r'=1 \\ l_{ir'} \neq -1}}^{R_k} (I(l_{ir} = l_{ir'}) - I(l_{ir} \neq l_{ir'})),$$

$$(2)$$

where $D_{kr}$ is a dataset only containing all instances that have been labeled by $u_r$ in $D_k$. $R_k$ is the number of all workers in $D_k$. $I(\cdot)$ is an indicator function that outputs 1 if the condition in the parentheses is true and 0 otherwise.

As we can see, the more consistent labels with other remained workers, the higher the $LC_r$. At the same time, the fewer inconsistent labels with other remained workers, the lower the $LC_r$. So the higher $LC_r$ indicates a higher level of label consistency of the remained worker $u_r$ with other remained workers. What's more, the positive and negative values of $LC_r$ are also indicative. When $LC_r$ is positive, it indicates that the labels from the remained worker $u_r$ are generally consistent with those from other remained workers. Thus, the existence of worker $u_r$ has improved the label consistency of $D_k$. So, it is more likely that the worker $u_r$ should indeed be remained. On the contrary, if $LC_r$ is negative, it indicates that the existence of the worker $u_r$ has damaged the label consistency of $D_k$. So, it is more likely that the worker $u_r$ should actually be filtered out.

If all remained workers have a high level of label consistency with other remained workers, the fitness of this hypothesis should be high. So, for the hypothesis $\boldsymbol{h}_k$, we sum up the label consistency of each remained worker with other remained workers to calculate its fitness as follows:

$$f(\boldsymbol{h}_k) = \sum_{r=1}^{R_k} LC_r. \qquad (3)$$

It should be noted that, the fitness $f(\boldsymbol{h}_k)$ calculated by Eq. (3) may be negative, which is not hoped in the definition of the fitness function. So we restrict the minimum value of the fitness to 0. That is, when the calculated fitness is less than 0, we directly set it to 0. What's more, there may exist some hypotheses that cause certain instances to have no labels after worker filtering. Since not all label integration methods can handle this situation, we also directly set the fitness of these hypotheses to 0.

Now, GA can be performed to identify the low-quality workers who should be filtered out. Furthermore, to guarantee the global convergence of LCWF, we have also adopted the widely used elitism strategy [Ahn and Ramakrishna, 2003] based on the standard GA. According to the elitism strategy, the best hypothesis in the current population will be copied directly into the next generation to replace the worst hypothesis, without any evolutionary operation.

Finally, when the number of generations reaches the maximum number of generations $T$, we can obtain an optimal

---

**Algorithm 1** LCWF $(D, f, K, T, s, m)$

---

**Input:** $D$: A crowdsourced dataset.
  $f$: A fitness function.
  $K, T, s, m$: The predefined parameters.

1: **Initialize**: Randomly generate $K$ hypotheses to $P$.
2: **Evaluate**: For each $\boldsymbol{h}_k$ in $P$, compute $f(\boldsymbol{h}_k)$ by Eq. (3).

3: **for** $t = 1$ to $T$ **do**
4:   Create a new generation $P^t$:
5:   **Select**: Select $(1-s) * K$ members of $P$ to $P^t$ according to the Eq. (1).
6:   **Crossover**: Select $s * K/2$ pairs of hypotheses from $P$ according to the Eq. (1). For each pair of hypotheses, produce two offspring and add them to $P^t$.
7:   **Mutate**: Randomly select $m$ percent of hypotheses from $P^t$ to mutate.
8:   **Update**: Find the best hypothesis $\boldsymbol{h}_b$ from $P$ to replace the worst hypothesis $\boldsymbol{h}_w$ from $P^t$. Then replace $P$ with $P^t$.
9:   **Evaluate**: For each $\boldsymbol{h}_k$ in $P$, compute $f(\boldsymbol{h}_k)$ by Eq. (3).
10: **end for**
11: Find the best hypothesis $\boldsymbol{h}_b$ from $P$.
12: Filter out labels from $D$ according to $\boldsymbol{h}_b$ to obtain $\hat{D}$.
13: Return $\hat{D}$.

---

solution to the worker filtering problem. According to this optimal solution, we filter out all labels of low-quality workers from $D$ to obtain $\hat{D}$, which can be fed into existing label integration methods for better performance. The whole learning process of LCWF is described in Algorithm 1.

## 4 EXPERIMENTS AND RESULTS

### 4.1 EXPERIMENTS SETUP

In this section, we evaluate the effectiveness of LCWF on improving the integration accuracy of existing label integration methods. We use the integration accuracy as the evaluation metric. Here, the integration accuracy is the proportion of integrated labels that are consistent with true labels. In our experiments, we use six state-of-the-art label integration methods: MV [Sheng et al., 2008], GTIC [Zhang et al., 2016], DEWSMV [Tao et al., 2021], MNLDP [Jiang et al., 2022], LAWMV [Chen et al., 2022c] and AALI [Zhang et al., 2023b]. We implement these label integration methods on the crowd environment and its knowledge analysis (CEKA) [Zhang et al., 2015] platform and keep the parameters of them consistent with the original papers. We also implement LCWF on the CEKA platform and set the four parameters $K, T, s, m$ to 80, 50, 0.5, 0.05, respectively. We repeat each experiment in this section 10 times independently, and report the averages as the final results.

Table 4: Integration accuracy (%) comparisons on the uniform distribution.

| Dataset | MV before | MV after | GTIC before | GTIC after | DEWSMV before | DEWSMV after | MNLDP before | MNLDP after | LAWMV before | LAWMV after | AALI before | AALI after |
|---|---|---|---|---|---|---|---|---|---|---|---|---|
| anneal | 88.57 | 92.22 ● | 55.79 | 74.25 ● | 88.36 | 92.05 ● | 91.92 | 94.52 ● | 82.24 | 87.37 ● | 90.04 | 93.64 ● |
| audiology | 78.81 | 84.51 ● | 76.64 | 79.73 | 78.98 | 84.56 ● | 76.73 | 83.45 ● | 78.14 | 84.87 ● | 80.84 | 83.27 ● |
| autos | 92.78 | 93.66 | 87.12 | 85.76 | 92.05 | 93.80 | 89.37 | 89.61 | 92.49 | 94.20 ● | 92.63 | 93.51 |
| balance-scale | 82.66 | 88.96 ● | 82.72 | 89.26 ● | 82.59 | 88.78 ● | 88.37 | 92.93 | 90.51 | 91.26 | 83.42 | 90.48 |
| biodeg | 77.39 | 80.78 ● | 77.39 | 82.32 ● | 77.21 | 81.08 ● | 83.08 | 86.53 ● | 89.54 | 90.65 ● | 81.35 | 84.85 ● |
| breast-cancer | 72.24 | 73.78 ● | 72.24 | 76.19 ● | 72.06 | 74.06 | 72.41 | 73.29 ● | 73.01 | 73.01 | 73.71 | 77.24 ● |
| breast-w | 76.92 | 79.11 | 76.92 | 80.80 ● | 76.77 | 79.37 | 89.81 | 91.97 | 95.65 | 95.74 | 78.11 | 86.71 ● |
| car | 80.08 | 87.82 ● | 77.32 | 88.33 ● | 80.02 | 87.64 ● | 82.69 | 88.44 ● | 78.39 | 86.20 ● | 81.35 | 88.52 ● |
| credit-a | 72.49 | 74.41 | 72.49 | 74.10 | 72.23 | 74.59 | 75.42 | 76.03 | 81.29 | 81.23 | 75.52 | 78.01 |
| credit-g | 75.28 | 79.77 ● | 75.28 | 82.23 ● | 75.22 | 79.92 ● | 73.46 | 76.95 ● | 76.99 | 78.62 ● | 76.28 | 82.75 ● |
| heart-c | 74.06 | 74.59 | 41.22 | 42.97 ● | 73.86 | 75.08 | 76.77 | 77.82 | 79.37 | 79.57 | 74.16 | 75.54 |
| heart-h | 77.28 | 81.63 ● | 46.16 | 49.25 ● | 76.94 | 80.95 ● | 80.27 | 83.64 ● | 85.07 | 86.46 ● | 77.35 | 81.29 ● |
| heart-statlog | 80.74 | 84.81 | 80.74 | 85.63 ● | 80.41 | 84.44 | 82.11 | 84.11 | 84.19 | 84.89 ● | 84.30 | 89.11 |
| hepatitis | 68.32 | 72.00 | 68.32 | 65.81 | 68.32 | 73.16 | 75.29 | 78.13 | 76.52 | 76.97 | 58.84 | 69.87 |
| horse-colic | 69.35 | 71.74 | 69.35 | 71.03 | 68.97 | 71.52 | 72.17 | 74.86 | 81.74 | 82.69 | 68.13 | 70.71 |
| hypothyroid | 85.32 | 90.78 ● | 67.29 | 81.18 ● | 85.19 | 90.73 ● | 94.11 | 96.13 ● | 92.29 | 92.29 ● | 85.54 | 90.63 ● |
| ionosphere | 71.48 | 72.91 ● | 71.48 | 71.68 | 71.51 | 72.88 | 78.38 | 76.72 ○ | 72.17 | 70.97 | 70.83 | 71.51 |
| diabetes | 81.98 | 86.25 | 81.98 | 87.64 | 81.61 | 85.72 | 81.32 | 85.18 | 81.61 | 82.76 | 83.41 | 88.91 |
| iris | 81.80 | 84.80 | 81.33 | 83.87 | 81.93 | 84.07 | 95.93 | 97.47 | 97.93 | 98.8 | 89.07 | 90.40 |
| kr-vs-kp | 65.89 | 65.98 | 65.89 | 66.17 | 65.68 | 65.82 | 75.09 | 73.74 ○ | 77.13 | 75.94 ○ | 69.95 | 69.52 |
| labor | 67.02 | 70.70 ● | 67.02 | 68.07 | 67.02 | 70.88 ● | 80.18 | 78.42 ○ | 79.47 | 78.95 ● | 74.74 | 74.21 ○ |
| letter | 97.34 | 96.23 | 97.56 | 96.99 | 97.34 | 96.78 | 98.54 | 99.67 | 98.45 | 99.89 | 98.56 | 99.45 |
| lymph | 76.22 | 85.54 ● | 75.81 | 86.42 ● | 75.74 | 85.27 ● | 81.15 | 89.80 ● | 84.86 | 90.27 ● | 77.30 | 87.23 ● |
| mushroom | 75.30 | 77.92 ● | 76.20 | 78.40 ● | 75.80 | 78.30 ● | 91.19 | 91.88 ● | 91.80 | 92.30 ● | 70.20 | 74.10 ● |
| segment | 94.62 | 92.23 ○ | 94.58 | 92.18 ○ | 94.55 | 92.14 ○ | 98.71 | 98.48 ○ | 96.23 | 97.35 | 96.01 | 94.96 ○ |
| sick | 70.49 | 73.04 ● | 70.49 | 71.51 ● | 70.16 | 72.73 ● | 81.96 | 83.99 ● | 91.77 | 91.77 | 71.87 | 75.17 ● |
| sonar | 68.08 | 71.49 ● | 68.08 | 71.35 ● | 68.22 | 71.01 ● | 76.88 | 81.59 ● | 76.88 | 78.75 ● | 68.94 | 73.08 ● |
| spambase | 72.44 | 75.35 ● | 72.44 | 75.66 ● | 72.11 | 75.20 ● | 77.66 | 79.74 ● | 80.02 | 80.74 ● | 75.46 | 79.12 ● |
| tic-tac-toe | 73.36 | 76.97 ● | 73.36 | 77.84 ● | 73.15 | 76.52 ● | 71.24 | 72.21 | 77.97 | 78.95 | 74.81 | 78.90 ● |
| vehicle | 92.15 | 92.96 | 92.14 | 92.73 | 92.13 | 92.98 | 92.55 | 92.71 | 90.30 | 93.10 | 93.43 | 94.16 |
| vote | 77.70 | 82.55 ● | 77.70 | 83.17 ● | 77.56 | 82.39 ● | 87.31 | 91.10 ● | 91.52 | 92.85 | 78.00 | 83.98 ● |
| vowel | 97.36 | 95.36 ○ | 97.49 | 95.34 ○ | 97.35 | 95.31 ○ | 99.94 | 99.79 | 97.93 | 97.08 ○ | 98.52 | 96.63 ○ |
| waveform | 90.04 | 93.02 ● | 90.03 | 92.99 | 89.79 | 93.05 ● | 94.6 | 95.72 | 97.82 | 98.25 | 92.31 | 95.23 |
| zoo | 86.83 | 87.43 ● | 89.90 | 88.81 | 87.13 | 87.03 | 93.56 | 95.15 | 93.56 | 96.24 | 89.80 | 92.48 |
| **Average** | 79.12 | 82.10 | 75.60 | 79.11 | 79.06 | 82.05 | 84.12 | 86.23 | 85.73 | 87.09 | 80.43 | 83.98 |
| **W/T/L** | - | 20/12/2 | - | 18/14/2 | - | 17/15/2 | - | 14/16/4 | - | 13/18/3 | - | 16/15/3 |

## 4.2 EXPERIMENTS ON SIMULATED DATA

We conduct our experiments on the whole 34 simulated crowdsourced datasets published on the CEKA platform[1]. Since some datasets have missing values and the competitor MNLDP [Jiang et al., 2022] used in our experiments cannot handle missing values, we use the mean of numeric features or the mode of nominal features from the available data to replace all missing feature values. After that, we simulate crowd workers to provide labels for instances. We first hide the true labels of all instances and then set the probability $p$ for each worker to provide labels consistent with the hidden true labels. In our simulated experiments, we set the number of workers to 9, and randomly generate each worker's label quality from a uniform distribution in the interval $[0.3, 0.9]$.

Table 4 shows the detailed integration accurcy comparison results of six label integration methods on 34 simulated crowdsourced datasets before and after worker filtering by LCWF. We also conduct corrected paired two-tailed t-tests [Nadeau and Bengio, 2003, Jiang et al., 2019b] at 95 percent significance level to compare the integration accuracy before and after worker filtering. The symbols ● and ○ in the table denote statistically improvement or degradation over its competitor, respectively. The average integration accuracy and the *Win/Tie/Lose (W/T/L)* values are summarized at the bottom of the table. We can see a significant improvement in the integration accuracy of these six methods on most of simulated datasets. Specifically, before worker filtering, the average integration accuracies of MV, GTIC, DEWSMV, MNLDP, LAWMV and AALI are 79.12%, 75.60%, 79.06%, 84.12%, 85.73% and 80.43%, respectively. After worker filtering, the integration accuracies of them are improved

[1]https://ceka.sourceforge.net

Table 5: Integration accuracy (%) comparisons on the Gaussian distribution.

| | MV | | GTIC | | DEWSMV | | MNLDP | | LAWMV | | AALI | |
|---|---|---|---|---|---|---|---|---|---|---|---|---|
| Dataset | before | after | before | after | before | after | before | after | before | after | before | after |
| anneal | 89.30 | 91.15 ● | 53.75 | 63.50 ● | 89.16 | 91.33 ● | 92.15 | 93.88 ● | 82.97 | 87.09 ● | 90.7 | 92.05 ● |
| audiology | 81.15 | 85.75 ● | 76.15 | 80.13 | 81.02 | 84.60 ● | 78.72 | 83.67 ● | 80.13 | 84.91 ● | 82.12 | 84.38 |
| autos | 92.05 | 92.49 | 85.56 | 86.98 | 91.8 | 92.00 | 88.63 | 88.54 | 91.61 | 93.56 | 91.37 | 93.07 |
| balance-scale | 83.6 | 87.89 ● | 83.73 | 88.13 ● | 83.39 | 87.81 ● | 88.75 | 92.43 ● | 90.34 | 90.67 | 84.75 | 90.88 ● |
| biodeg | 72.42 | 75.19 ● | 72.42 | 76.75 ● | 72.23 | 75.03 ● | 78.26 | 81.18 ● | 85.57 | 86.68 | 74.88 | 78.91 ● |
| breast-cancer | 68.88 | 74.16 ● | 67.48 | 74.62 ● | 68.74 | 74.06 ● | 70.59 | 74.51 ● | 75.03 | 76.64 | 72.73 | 77.24 ● |
| breast-w | 74.64 | 79.18 | 74.64 | 82.35 | 74.38 | 79.77 | 89.04 | 92.93 | 95.67 | 95.77 | 74.68 | 87.95 |
| car | 78.45 | 87.40 ● | 74.69 | 87.10 ● | 78.32 | 87.41 ● | 82.24 | 88.58 ● | 76.97 | 86.67 ● | 79.57 | 88.41 ● |
| credit-a | 70.83 | 75.10 ● | 70.83 | 74.30 ● | 70.59 | 75.09 ● | 75.28 | 77.32 ● | 82.17 | 82.64 ● | 72.43 | 77.03 ● |
| credit-g | 70.74 | 72.15 | 70.74 | 72.83 | 70.4 | 72.01 | 70.8 | 71.15 | 72.8 | 73.17 | 72.77 | 75.09 ● |
| heart-c | 70.1 | 75.05 ● | 36.67 | 39.57 ● | 70.00 | 75.25 ● | 73.86 | 78.61 ● | 81.49 | 83.66 ● | 70.13 | 76.20 ● |
| heart-h | 75.58 | 78.78 | 40.41 | 46.43 ● | 75.37 | 78.57 | 79.29 | 82.99 | 83.47 | 85.85 | 75.65 | 79.86 ● |
| heart-statlog | 73.44 | 76.74 ● | 73.44 | 77.04 ● | 73.37 | 76.89 ● | 76.15 | 78.63 | 79.85 | 80.37 | 77.67 | 81.33 ● |
| hepatitis | 72.39 | 75.23 | 71.16 | 73.74 | 72.65 | 74.71 | 78.58 | 78.9 | 83.03 | 79.81 | 68.9 | 71.29 |
| horse-colic | 71.33 | 73.56 | 71.33 | 72.66 | 71.09 | 73.29 | 75.24 | 75.49 | 84.59 | 80.87 | 68.59 | 70.6 |
| hypothyroid | 83.21 | 87.49 ● | 64.12 | 77.21 ● | 83.21 | 87.51 ● | 92.9 | 95.29 ● | 92.29 | 92.30 ● | 83.50 | 87.69 ● |
| ionosphere | 79.06 | 82.22 ● | 79.06 | 81.57 | 78.58 | 82.22 ● | 86.89 | 88.09 | 80.60 | 81.79 | 81.57 | 82.79 |
| diabetes | 67.62 | 70.7 | 67.62 | 72.34 | 67.55 | 70.22 | 71.00 | 72.72 | 73.2 | 74.18 | 70.39 | 74.73 |
| iris | 87.6 | 91.13 | 86.93 | 90.87 | 87.6 | 91.4 | 98.13 | 98.6 | 98.27 | 99.00 | 93.27 | 96.6 |
| kr-vs-kp | 76.37 | 78.59 | 76.37 | 78.8 | 76.11 | 78.33 | 85.21 | 84.85 ○ | 87.07 | 85.64 ○ | 80.07 | 81.79 |
| labor | 72.81 | 74.21 | 72.81 | 71.93 ○ | 72.63 | 73.51 | 78.95 | 82.81 ● | 83.16 | 86.32 ● | 76.49 | 77.19 |
| letter | 97.78 | 96.9 | 97.89 | 96.83 | 97.82 | 96.98 | 98.56 | 99.78 ● | 98.56 | 99.65 ● | 98.12 | 99.53 ● |
| lymph | 79.05 | 86.22 ● | 80.14 | 86.01 ● | 79.39 | 86.22 ● | 83.65 | 88.85 ● | 85.41 | 90.20 ● | 80.68 | 86.55 ● |
| mushroom | 75.79 | 77.98 ● | 75.79 | 78.07 ● | 75.49 | 77.88 ● | 91.28 | 91.71 ● | 92.4 | 92.73 ● | 69.5 | 73.90 ● |
| segment | 96.58 | 95.34 ○ | 96.52 | 95.41 ○ | 96.6 | 95.55 ○ | 99.05 | 98.92 ○ | 96.75 | 97.60 | 97.83 | 97.47 ○ |
| sick | 77.57 | 80.16 ● | 77.57 | 80.33 ● | 77.31 | 79.92 | 89.37 | 90.79 ● | 93.85 | 93.85 ● | 79.12 | 81.03 ● |
| sonar | 70.38 | 74.57 ● | 70.38 | 74.18 ● | 70.19 | 74.76 ● | 77.74 | 81.59 ● | 77.69 | 81.20 ● | 73.08 | 75.82 ● |
| spambase | 70.23 | 73.74 ● | 70.23 | 74.81 ● | 69.94 | 73.61 ● | 76.98 | 80.80 ● | 82.06 | 82.76 | 71.91 | 78.28 ● |
| tic-tac-toe | 74.75 | 78.86 ● | 74.75 | 80.11 ● | 74.58 | 78.76 ● | 71.57 | 72.16 | 80.76 | 82.53 | 76.02 | 81.46 |
| vehicle | 89.54 | 90.92 | 89.63 | 90.78 | 89.14 | 90.78 | 91.97 | 91.77 | 88.58 | 91.65 | 90.93 | 91.87 |
| vote | 70.94 | 75.38 ● | 70.94 | 75.63 ● | 70.87 | 74.28 ● | 80.23 | 82.48 | 83.52 | 83.77 ● | 72.14 | 75.45 |
| vowel | 96.57 | 92.92 ○ | 96.68 | 93.02 ○ | 96.48 | 92.8 ○ | 99.92 | 99.54 ○ | 96.54 | 96.04 ○ | 97.96 | 94.34 ○ |
| waveform | 86.06 | 90.35 ● | 86.01 | 90.38 | 85.83 | 90.29 ● | 93.08 | 94.69 | 96.96 | 97.35 | 90.03 | 92.82 |
| zoo | 86.93 | 87.72 ● | 88.42 | 89.7 | 87.13 | 88.42 | 92.28 | 94.16 | 93.66 | 95.25 | 90.79 | 92.18 |
| Average | 78.93 | 81.92 | 74.85 | 78.65 | 78.79 | 81.80 | 84.01 | 86.13 | 86.09 | 87.42 | 80.30 | 83.70 |
| W/T/L | - | 20/12/2 | - | 17/15/2 | - | 18/14/2 | - | 16/15/3 | - | 13/19/2 | - | 17/14/3 |

to 82.10%, 79.11%, 82.05%, 86.23%, 87.09% and 83.98%, respectively.These results powerfully demonstrate the effectiveness of LCWF on improving integration accuracy.

Besides, to verify the effectiveness of LCWF for different labeling quality distributions, we conduct another experiment, in which we randomly generate the labeling quality from a Gaussian distribution with $N(0.6, 0.3^2)$. Table 5 shows the detailed experimental results of this experiment. Before worker filtering, the average integration accuracies of MV, GTIC, DEWSMV, MNLDP, LAWMV and AALI are 78.93%, 74.85%, 78.79%, 84.01%, 86.09% and 80.30%, respectively. After worker filtering, the integration accuracies of them are improved to 81.92%, 78.65%, 81.80%, 86.13%, 87.42% and 83.70%, respectively. Thus, we can draw to the conclusion that whether the simulated labeling quality of

crowd worker belongs to a uniform or Gaussian distribution, LCWF can notably improve the integration accuracy of existing label integration methods.

Finally, to more directly demonstrate the effectiveness of LCWF, we also conduct the experiments on the crowd label quality before and after worker filtering using LCWF both on the uniform and Gaussian distribution. Here, the crowd label quality is the proportion of crowd labels that are consistent with true labels in the dataset. Table 6 shows the detailed results of these experiments. After worker filtering, the average crowd label quality can be improved from 59.58 % to 66.16% on the uniform distribution and the average crowd label quality can be improved from 59.47% to 66.02% on the Gaussian distribution. All these experimental results powerfully demonstrate the effectiveness of LCWF again.

Table 6: Crowd label quality (%) comparisons on the uniform distribution and Gaussian distribution.

| | uniform | | Gaussian | |
|---|---|---|---|---|
| Dataset | before | after | before | after |
| anneal | 59.29 | 71.21 | 61.14 | 70.66 |
| audiology | 53.8 | 67.12 | 56.21 | 66.81 |
| autos | 60.39 | 72.32 | 59.29 | 72.7 |
| balance-scale | 61.03 | 69.57 | 62.00 | 68.8 |
| biodeg | 61.51 | 65.17 | 59.32 | 61.91 |
| breast-cancer | 59.11 | 62.00 | 58.17 | 61.94 |
| breast-w | 61.8 | 64.24 | 60.29 | 64.98 |
| car | 58.63 | 68.66 | 57.33 | 69.75 |
| credit-a | 60.54 | 63.09 | 58.65 | 63.03 |
| credit-g | 60.86 | 65.24 | 58.75 | 60.91 |
| heart-c | 60.68 | 62.42 | 58.22 | 62.13 |
| heart-h | 61.90 | 66.18 | 61.33 | 64.06 |
| heart-statlog | 62.65 | 68.11 | 59.99 | 63.5 |
| hepatitis | 57.38 | 61.13 | 60.43 | 63.79 |
| horse-colic | 58.22 | 60.06 | 58.73 | 61.71 |
| hypothyroid | 60.11 | 71.77 | 58.92 | 68.05 |
| ionosphere | 59.29 | 61.31 | 62.26 | 65.68 |
| diabetes | 64.08 | 68.56 | 57.63 | 60.42 |
| iris | 56.44 | 67.75 | 60.11 | 71.28 |
| kr-vs-kp | 56.66 | 57.24 | 61.21 | 64.32 |
| labor | 58.01 | 60.18 | 59.69 | 62.75 |
| letter | 58.67 | 71.56 | 58.89 | 70.56 |
| lymph | 58.43 | 68.11 | 60.44 | 67.49 |
| mushroom | 61.20 | 64.40 | 61.41 | 64.05 |
| segment | 57.82 | 73.16 | 60.54 | 73.4 |
| sick | 58.57 | 60.70 | 62.21 | 64.61 |
| sonar | 57.29 | 60.33 | 58.94 | 62.32 |
| spambase | 59.85 | 62.67 | 58.5 | 61.54 |
| tic-tac-toe | 59.49 | 62.81 | 60.34 | 64.43 |
| vehicle | 60.46 | 71.37 | 57.22 | 71.23 |
| vote | 62.36 | 65.34 | 58.54 | 62.94 |
| vowel | 60.41 | 73.82 | 58.33 | 71.55 |
| waveform | 62.26 | 72.77 | 58.55 | 70.43 |
| zoo | 56.62 | 68.9 | 58.17 | 70.77 |
| **Average** | 59.58 | 66.16 | 59.47 | 66.02 |

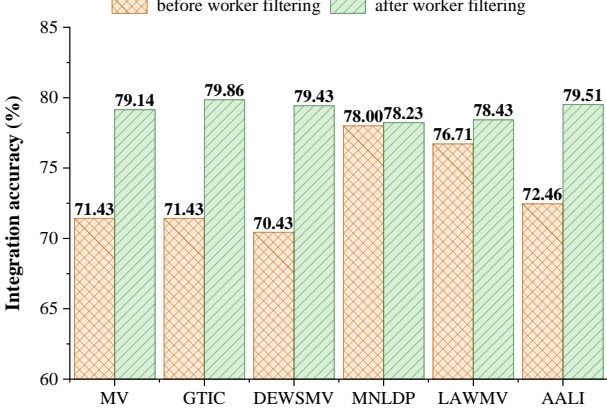

Figure 1: Integration accuracy (%) comparisons before and after worker filtering by LCWF.

## 4.3 EXPERIMENTS ON REAL-WORLD DATA

To further valid the effectiveness of LCWF, we also conduct our experiments on the real-world crowdsourced dataset "Music_genre" [Rodrigues et al., 2013], which is collected from Amazon Mechanical Turk (AMT) platform. The "Music_genre" dataset is a classic multi-class crowdsourced dataset that contains 700 instances, which are described by 124 features and 44 crowd workers are employed to label these instances. Totally, the dataset gets 2946 crowd labels, but nearly 43.93% of the crowd labels are noise, which is relatively high. Thus, it is suitable for LCWF to purify noisy labels, that's why we select this dataset for experiments.

Figure 1 shows the integration accuracy comparison results of six methods before and after worker filtering by LCWF. We can see a significant improvement in the integration ac-

curacy of these six methods. Specifically, before worker filtering, the integration accuracies of MV, GTIC, DEWSMV, MNLDP, LAWMV and AALI are 71.43%, 71.43%, 70.43%, 78.00%, 76.71% and 72.46 %, respectively. After worker filtering, the integration accuracies of them are improved to 79.14%, 79.86%, 79.43%, 78.23%, 78.43% and 79.51%, respectively. The comparison results on the real-world dataset powerfully demonstrate the effectiveness of LCWF on improving integration accuracy again.

## 4.4 PARAMETER SENSITIVITY ANALYSIS OF LCWF

LCWF has four parameters: The fraction of the population to be replaced by crossover $s$, the mutation rate $m$, the population size $K$ and the maximum of generations $T$. To analyse the influence of different settings of these four parameters on LCWF, we conduct following two series of experiments on the dataset "Music_genre". In these experiments, we use the crowd label quality as the evaluation metric.

1. In the first experiment, we fix the value of $s$ and $m$ at 0.5 and 0.05, respectively, which refer to the settings in the paper [Jiang et al., 2005]. And then, we analyze the influence of $K$ and $T$ on the performance of LCWF.

2. In the second experiment, we fix the value of $K$ and $T$ at 80 and 50, respectively. And then, we analyze the influence of $s$ and $m$ on the performance of LCWF.

In the first experiment, we first fix $T$ at 50 and observe the performance of LCWF as $K$ increases from 10 to 100. Figure 2 (a) shows the detailed comparison results. We can see that when $K$ increases from 10 to 80, the crowd label quality improves. When $K$ is larger than 80, the crowd label quality varies slightly and tends to converge. This indicates that, the performance of LCWF is not sensitive to $K$ when

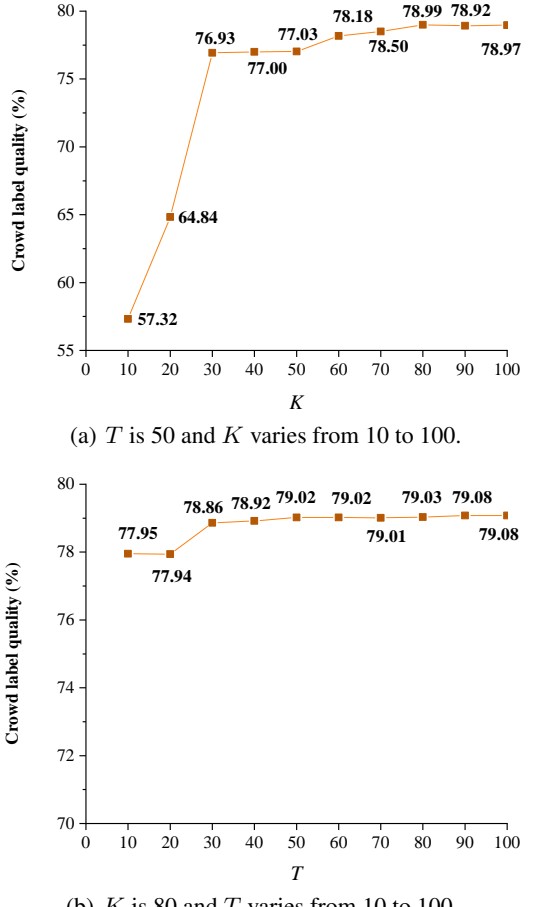

(a) $T$ is 50 and $K$ varies from 10 to 100.

(b) $K$ is 80 and $T$ varies from 10 to 100.

Figure 2: Crowd label quality comparisons on different settings of $K$ and $T$ when $s$ and $m$ are fixed at 0.5 and 0.05.

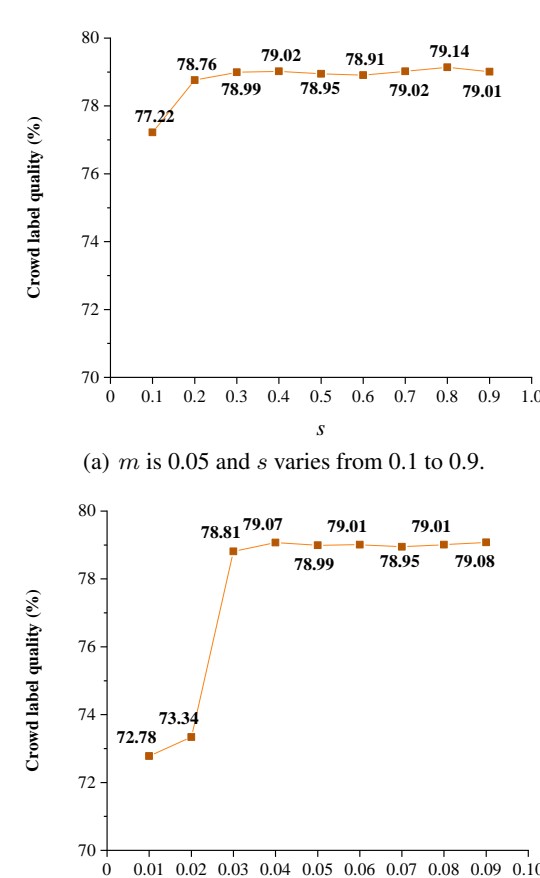

(a) $m$ is 0.05 and $s$ varies from 0.1 to 0.9.

(b) $s$ is 0.5 and $m$ varies from 0.01 to 0.09.

Figure 3: Crowd label quality comparisons on different settings of $s$ and $m$ when $K$ and $T$ are fixed at 80 and 50.

$K$ is around or larger than 80. Then, we fix $K$ at 80 and observe the performance of LCWF when $T$ increases from 10 to 100. Figure 2 (b) shows the detailed comparison results. We can see that the performance of LCWF is not sensitive to $T$ when $T$ is around or larger than 50.

In the second experiment, we first fix $m$ at 0.05 and observe the performance as $s$ increases from 0.1 to 1. Figure 3 (a) shows the detailed comparison results. We can see that the performance of LCWF is not sensitive to $s$ when $s$ is around or larger then 0.2. Then, we fix the parameter $s$ at 0.5 and observe the performance when $m$ increases from 0.01 to 0.1. Figure 3 (b) shows the detailed comparison results. We can see that the performance of LCWF is not sensitive to $m$ when $m$ is around or larger than 0.03.

Based on above experimental results, we set $K$, $T$, $s$ and $m$ to 80, 50, 0.5, 0.05, respectively, in our experiments. From Figure 2 and Figure 3, we can see that around this parameter settings, the performance of LCWF is not sensitive.

## 4.5 DISCUSSION

Based on above experiment results on the simulated and real-world datasets, our proposed LCWF can perform well on improving the integration accuracy of the existing label methods. Although we have demonstrated the effectiveness of LCWF, it is actually not solid for every scenarios. For example, when most workers show low-quality, then most labels annotated on the same instances may be incorrect. Thus, high-quality workers can hardly reach consensus with other low-quality workers. In this case, if a high-quality worker is filtered out, the label consistency of the multiple noisy labels may even improve. Based on the current fitness function, the hypotheses which filter out high-quality workers can have a less fitness than the hypotheses which filter out low-quality workers, which will lead to poor performance on filtering out low-quality workers.

What' more, there may exist collusion between workers, it means that some workers provide totally consistent labels on most or even all of instances they have annotated. If

the consistent labels of these collusive workers are correct, then worker collusion can be beneficial for LCWF to filter out low-quality workers. However, if the consistent labels of these collusive workers are incorrect, then LCWF may mistake these workers as high-quality workers due to the high label consistency of their labels and filter out the actual high-quality workers instead. Thus, the performance of LCWF is not stable when there is worker collusion.

Since these scenarios we discuss above are not common in the crowdsourcing and most of the label integration methods do not consider these special scenarios, we do not consider them in the current version of LCWF too. In these scenarios, the essential reason why LCWF cannot work well is that the label consistency with other workers become unreliable for the identification of low-quality workers. Therefore, to adapt to these special scenarios, the effort can be made to relax the dependence on the other workers, which is one of the directions to further improve or extend LCWF.

## 5 CONCLUSIONS AND FUTURE WORK

In this paper, we proposes a label consistency-based worker filtering (LCWF) algorithm. Different from most existing works focusing on designing more complex label integration methods, LCWF aim to purify noisy labels for existing label integration methods, which provides a new perspective for improving the integration performance. In LCWF, we transform the worker filtering problem into a discrete optimization problem and utilize label consistency to define the fitness function. Then, a genetic algorithm is used to search for the optimal solution. Experimental results on real-world and simulated datasets show that LCWF can effectively purify noisy labels and improve the integration accuracy of existing label integration methods.

In the current version of LCWF, we identify low-quality workers and then filter out all labels from them. Actually, the label quality of workers should be instance-dependent. Even low-quality workers can also provide high-quality labels on certain instances, so filtering out the low-quality workers directly may cause the loss of correct labels. Therefore, we believe that extending LCWF to be instance-dependent worker filtering can further improve its performance. What's more, the current version of LCWF cannot deal with some special scenarios, such as high proportion of low quality workers and worker collusion, which is also an important extension direction for us in the future.

### Acknowledgements

We thank the anonymous reviewers for their helpful comments. The work was supported by National Natural Science Foundation of China (62276241).

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
