# OpenReview forum: "Label Consistency-based Worker Filtering for Crowdsourcing"
_auai.org/UAI/2024/Conference — UAI 2024 poster_

### Official Review · Reviewer_gz4H · 2024-03-18

**Q2-1 Originality-Novelty:** 4
**Q2-2 Correctness-Technical Quality:** 3
**Q2-5 Clarity Of Writing:** 3

**Q1 Summary And Contributions:**

This paper proposes a new worker filtering method LCWF in the context of crowdsourcing. The main innovation is to filter unreliable workers based on consistency before label integration, thereby purifying noise labels. LCWF transforms the problem of worker filtering into a discrete optimization problem, uses genetic algorithms to find the optimal solution to the problem, and uses label consistency to define a fitness function. LCWF was performed on the 34 simulated crowdsourcing datasets and the Music_genre dataset. The integration results on these processed datasets generated by multiple previous label integration algorithms were significantly improved.

**Q2-3 Extent To Which Claims Are Supported By Evidence:**

4: Excellent: all claims are supported by very convincing evidence (in the form of comprehensive experimental evaluation, rigorous mathematical proofs, detailed (pseudo-)code, precise references, well-motivated and realistic assumptions) and the authors deliver what they promise.

**Q2-4 Reproducibility:**

4: Excellent: key resources (e.g. proofs, code, data) are available and key details (e.g. proof sketches, experimental setup) are comprehensively described for competent researchers to confidently and easily reproduce the main results.

**Q3 Main Strengths:**

1. This paper is easy to understand and proposes a strategy for filtering unreliable workers based on consistency before label integration, without making specific assumptions about the labeling strategy for unreliable workers.  Compared to previous methods, it can be applied to multiple classification problems.
2. On the premise that the consistency of labels with other workers is reliable for identifying low-quality workers, various label integration algorithms have achieved good improvements in the results of running on LCWF-filtered datasets.
3. The authors summited the source code and datasets, which show the proposed method is reproducible. The experiments are comprehensive and convincing.

**Q4 Main Weakness:**

1. LCWF filters unreliable workers based on consistency, thereby purifying noise labels. The paper only provides a comparison of the results of running label integration algorithms on the labeled datasets before and after LCWF processing. It would be better if the authors could provide direct comparison experiments on the quality of group labels before and after using LCWF.
2. In the experiment on the Music_genre dataset, the authors explained that LCWF is insensitive to the threshold values of parameters s and m. When selecting the values of s and m, they referred to the paper "Evolutional naive Bayes. In Proceedings of the 1st International Symposium on Intelligence Computation and Applications."  The authors should provide a more detailed explanation of the settings of these parameters.

**Q5 Detailed Comments To The Authors:**

This paper proposes a new worker filtering method LCWF in the context of crowdsourcing. The main innovation is to filter unreliable workers based on consistency before label integration, thereby purifying noise labels. LCWF transforms the problem of worker filtering into a discrete optimization problem, uses genetic algorithms to find the optimal solution to the problem, and uses label consistency to define a fitness function. LCWF was performed on the 34 simulated crowdsourcing datasets and the Music_genre dataset. The integration results on these processed datasets generated by multiple previous label integration algorithms were significantly improved.

The key innovation of the paper is transforming the problem of worker filtering into a discrete optimization problem and using genetic algorithms to find the optimal solution to the problem. The work first shows that genetic algorithms can be applied to crowdsourcing worker filtering. It provides another train of thought to this difficult traditional problem in crowdsourcing learning. Besides, there are also some merits that are listed in Q2. Also, some minor weaknesses listed in Q3 can be easily addressed in the camera-ready version if the paper can be accepted. Overall, the paper is well-written and should be accepted at the conference.

**Q9 Complying With Reviewing Instructions:**

Yes

---

> ### Author Rebuttal · Authors · 2024-04-06
>
> Reviewer gz4H
> Q1: LCWF filters unreliable workers based on consistency, thereby purifying noise labels. The paper only provides a comparison of the results of running label integration algorithms on the labeled datasets before and after LCWF processing. It would be better if the authors could provide direct comparison experiments on the quality of group labels before and after using LCWF.
>
> Author Response:
> Thanks for your valuable comments. Indeed, the comparison experiments on the crowd label quality before and after worker filtering using LCWF is more direct for verifying the effectiveness of LCWF. Actually, we have conducted the crowd label quality comparison experiments on 34 simulated datasets both on the uniform distribution and Gaussian distribution. Due to the page limitation, we do not exhibit the experimental results in section 4.2. However, we have provided the experimental results in the appendix, which are shown in Table 6. From Table 6, we can see that regardless of uniform or Gaussian distribution, LCWF can effectively filter noisy labels and improve the crowd label quality, which further demonstrate the effectiveness of LCWF. In the final version, we will enhance the introduction of these comparison experiments to more powerfully demonstrate the effectiveness of LCWF. Thanks again for your valuable comments.
>
> Reviewer gz4H
> Q2: In the experiment on the Music_genre dataset, the authors explained that LCWF is insensitive to the threshold values of parameters s and m. When selecting the values of s and m, they referred to the paper "Evolutional naive Bayes. In Proceedings of the 1st International Symposium on Intelligence Computation and Applications." The authors should provide a more detailed explanation of the settings of these parameters.
>
> Author Response:
> Thanks for your valuable comments. Since LCWF has totally four parameters (s, m , K, T), it is challenging to conduct the sensitivity analysis experiments for all these four parameters simultaneously. Therefore, we analyse the sensitivity of these four parameters by two experiments. In the first experiment, we fix the parameters s, m by referring to the setup in the paper "Evolutional naive Bayes." to conduct the sensitivity analysis experiments for the parameters K and T, the results can be seen in Figure 2 (a) and (b). Based on the results of the first experiment, we then fix the parameters K and T to conduct the sensitivity analysis experiments for the parameters s and m, the results can be seen in Figure 3 (a) and (b). From Figure 2 and Figure 3, we can see that when the parameters s, m, K, T are set around 0.5, 0.05, 80, 50, the performance of LCWF is insensitive, that’s why we adopt this parameter setting in all experiments. In the final version, we will enhance the explanation of the settings of these parameters based on the results of sensitivity analysis experiments. Thanks again for your valuable comments.

---

### Official Review · Reviewer_14Gc · 2024-03-22

**Q2-1 Originality-Novelty:** 3
**Q2-2 Correctness-Technical Quality:** 3
**Q2-5 Clarity Of Writing:** 4

**Q10 Ethical Concerns:**

No.

**Q1 Summary And Contributions:**

Different from existing works focusing on designing more complex label integration methods to infer unknown true labels more accurately from multiple noisy labels, this paper aims to purify noisy labels for existing label integration methods and proposes a label consistency-based worker filtering (LCWF) algorithm.

**Q2-3 Extent To Which Claims Are Supported By Evidence:**

3: Good: the main claims are supported by convincing evidence (in the form of adequate experimental evaluation, proofs, (pseudo-)code, references, assumptions).

**Q2-4 Reproducibility:**

4: Excellent: key resources (e.g. proofs, code, data) are available and key details (e.g. proof sketches, experimental setup) are comprehensively described for competent researchers to confidently and easily reproduce the main results.

**Q3 Main Strengths:**

S1. This paper proposes to purify noisy labels before label integration to infer unknown true labels more accurately, which provides a new perspective for improving the performance of label integration.
S2. This paper proposes a label consistency-based worker filtering (LCWF) algorithm to purify noisy labels for existing label integration methods, which transform the worker filtering problem into a discrete optimization problem and utilize label consistency to define the fitness function.
S3. Extensive experiments on real-world and simulated datasets show that LCWF can effectively purify noisy labels and improve the integration accuracy of existing label integration methods.

**Q4 Main Weakness:**

W1. Some contents of the motivation in Subsection 3.1 could be written into the Introduction. The motivation of GA based method should be clarified. In particular, the noisy label purification could be regarded as a prediction problem, and what’s the advantage of the authors’ work compared to the deep neural network based idea?

W2. I have some minor concerns about the experiments:
1) In Subsection 4.2, EXPERIMENTS ON SIMULATED DATA: With regard to the comparison results, statistical tests are needed in the comparison results.
2) In Subsection 4.2, EXPERIMENTS ON SIMULATED DATA: “Since some datasets have missing values and some label integration methods used in our experiments cannot handle missing values, we use the mean of numeric features or the mode of nominal features from the available data to replace all missing feature values”. Please specify the specific algorithms that cannot handle missing values.
3) In Subsection 4.3, EXPERIMENTS ON REAL-WORLD DATA: The real-world crowdsourced dataset “Music_genre" collected from Amazon Mechanical Turk (AMT) platform was used. Why is this dataset appropriate for evaluating the proposed algorithm? Some necessary justifications or clarifications should be given.

**Q5 Detailed Comments To The Authors:**

See the above weakness points.

**Q9 Complying With Reviewing Instructions:**

Yes

---

> ### Author Rebuttal · Authors · 2024-04-06
>
> Reviewer 14Gc
> Q1: In Subsection 4.2, EXPERIMENTS ON SIMULATED DATA: With regard to the comparison results, statistical tests are needed in the comparison results.
>
> Author Response:
> Thanks for your valuable comments. Indeed, statistical test is a convincing way to prove the effectiveness of our proposed method on improving the integration accuracy. In the final version, we will conduct the Wilcoxon signed-rank test or the corrected paired two-tailed t-test based on the comparison results of integration accuracy before and after worker filtering by LCWF. Thanks again for your valuable comments.
>
> Reviewer 14Gc
> Q2: In Subsection 4.2, EXPERIMENTS ON SIMULATED DATA: “Since some datasets have missing values and some label integration methods used in our experiments cannot handle missing values, we use the mean of numeric features or the mode of nominal features from the available data to replace all missing feature values”. Please specify the specific algorithms that cannot handle missing values.
>
> Author Response:
> Thanks for your valuable comments. Because the competitor MNLDP [1] used in our experiment cannot deal with missing values, we preprocess the dataset to replace all missing values before experiment. In the final version, we will specify that specific algorithm MNLDP that cannot handle missing values. Thanks again for your valuable comments.
> [1] Learning from Crowds with Multiple Noisy Label Distribution Propagation. IEEE Transactions on Neural Networks and Learning Systems, 2022, 33(11): 6558-6568.
>
> Reviewer 14Gc
> Q3: In Subsection 4.3, EXPERIMENTS ON REAL-WORLD DATA: The real-world crowdsourced dataset “Music_genre" collected from Amazon Mechanical Turk (AMT) platform was used. Why is this dataset appropriate for evaluating the proposed algorithm? Some necessary justifications or clarifications should be given.
>
> Author Response:
> Thanks for your valuable comments. In the experiments on real-world dataset, we use the dataset “Music_genre". In this dataset, we note that 43.93% of crowd labels are noise, which has a relatively low crowd label quality. Therefore, it is suitable for LCWF to purify noisy labels, that’s why we select this dataset for experiments. In the final version of paper, we will enhance the explanation for the dataset selection. Thanks again for your valuable comments.

---

### Official Review · Reviewer_3m9P · 2024-03-22

**Q2-1 Originality-Novelty:** 4
**Q2-2 Correctness-Technical Quality:** 4
**Q2-5 Clarity Of Writing:** 4

**Q1 Summary And Contributions:**

Rather than just designing more complex label integration methods to infer unknown true labels from multiple noisy labels, this paper proposes to purify noisy labels by worker filtering to improve the performance of existing label integration methods.

They first transform the worker filtering problem into a discrete optimization problem and utilize label consistency to define the fitness function for this problem. Then, they search for the optimal solution to this problem by a genetic algorithm. Finally, they filter out all labels from low-quality workers according to the optimal solution they obtained.

**Q2-3 Extent To Which Claims Are Supported By Evidence:**

3: Good: the main claims are supported by convincing evidence (in the form of adequate experimental evaluation, proofs, (pseudo-)code, references, assumptions).

**Q2-4 Reproducibility:**

3: Good: key resources (e.g. proofs, code, data) are available and key details (e.g. proofs, experimental setup) are sufficiently well-described for competent researchers to confidently reproduce the main results.

**Q3 Main Strengths:**

The main idea of the paper is novel as it is not to design more complex label integration methods, but purify noisy labels by worker filtering before label integration. This opens up a new perspective to research label integration problems.

The paper is technically sound. I have carefully checked the details.

The experiments are detailed. The proposed method has been compared with different label integration methods on simulated and real-world data sets. The experiment results support the main claim.

The paper is well organized and clearly written.

**Q4 Main Weakness:**

No obvious weakness.

**Q5 Detailed Comments To The Authors:**

One item is incorrectly colored in table3.

**Q9 Complying With Reviewing Instructions:**

Yes

---

> ### Author Rebuttal · Authors · 2024-04-06
>
> Reviewer 3m9P
> Q1: One item is incorrectly colored in table 3.
>
> Author Response:
> Thanks for your valuable comments. Indeed, there is one item is incorrectly colored as red in table 3. In the final version, we will fix this mistake.

---

### Official Review · Reviewer_G6bq · 2024-03-23

**Q2-1 Originality-Novelty:** 3
**Q2-2 Correctness-Technical Quality:** 3
**Q2-5 Clarity Of Writing:** 3

**Q1 Summary And Contributions:**

This paper addresses the issue of noisy labels in crowdsourcing scenarios and focuses on purifying the labels before integrating them. The proposed label consistency-based worker filtering (LCWF) algorithm aims to filter out low-quality workers based on the idea that high label consistency indicates high-quality workers. The algorithm transforms the worker filtering problem into a discrete optimization problem and utilizes label consistency as the fitness function. Through the use of a genetic algorithm, the optimal solution is found, and labels from low-quality workers are filtered out accordingly. Experimental results on simulated and real-world datasets demonstrate the effectiveness of LCWF in purifying noisy labels and improving the accuracy of label integration methods.

**Q2-3 Extent To Which Claims Are Supported By Evidence:**

3: Good: the main claims are supported by convincing evidence (in the form of adequate experimental evaluation, proofs, (pseudo-)code, references, assumptions).

**Q2-4 Reproducibility:**

3: Good: key resources (e.g. proofs, code, data) are available and key details (e.g. proofs, experimental setup) are sufficiently well-described for competent researchers to confidently reproduce the main results.

**Q3 Main Strengths:**

1. The paper introduces a fresh perspective by focusing on purifying noisy labels before the label integration process, which offers a new approach for improving label integration performance.
2. The proposed LCWF algorithm effectively filters out low-quality workers by leveraging label consistency, resulting in the removal of noisy labels associated with unreliable workers.

**Q4 Main Weakness:**

1. The current version of LCWF assumes that low-quality workers should be filtered out completely, without considering the potential variability in their label quality across different instances.  This approach may result in the loss of correct labels provided by low-quality workers for certain instances, potentially impacting the overall integration accuracy.
2. The paper lacks a comprehensive theoretical analysis of the proposed LCWF algorithm.  The absence of a deeper theoretical understanding, such as convergence guarantees or complexity analysis, hinders the assessment of the algorithm's robustness and scalability.

**Q5 Detailed Comments To The Authors:**

While the experimental results are promising, including a deeper theoretical analysis would strengthen the work.  Could you please consider providing insights into the convergence properties, computational complexity, or statistical guarantees of the proposed LCWF algorithm.  This would enhance the understanding of its underlying principles and provide a stronger foundation for its application.

**Q9 Complying With Reviewing Instructions:**

Yes

---

> ### Author Rebuttal · Authors · 2024-04-06
>
> Reviewer G6bq
> Q1: The current version of LCWF assumes that low-quality workers should be filtered out completely, without considering the potential variability in their label quality across different instances. This approach may result in the loss of correct labels provided by low-quality workers for certain instances, potentially impacting the overall integration accuracy.
>
> Author Response:
> Thanks for your valuable comments. Indeed, as we have discussed in the section 5, the label quality of workers should be instance-dependent. Even low-quality workers can also provide high-quality labels on certain instances, so filtering out the low-quality workers directly may cause the loss of correct labels. However, according to the extensive experimental results on simulation and real-world datasets, our LCWF can significantly improve the integration accuracy of existing label integration methods. This indicates that even though the loss of some correct labels, filtering low-quality workers is still beneficial. Consistent with your views, we also believe that extending LCWF to be instance-dependent worker filtering can further improve its performance, which is an important research direction for our future work and has been discussed in section 5. In the final version, we will enhance the explanation and discuss of this improvement direction of LCWF. Thanks again for your valuable comments.
>
> Reviewer G6bq
> Q2: The paper lacks a comprehensive theoretical analysis of the proposed LCWF algorithm. The absence of a deeper theoretical understanding, such as convergence guarantees or complexity analysis, hinders the assessment of the algorithm's robustness and scalability.
>
> Author Response:
> Thanks for your valuable comments. Indeed, the time complexity analysis can promote the assessment of LCWF's robustness and scalability. Assuming that the maximum number of generations is T , the population size is K, the length of the hypothesis is R, and the number of the instances in the crowdsourced dataset is N. The time complexity of LCWF in one evolutionary loop is analyzed as follows. For the three evolutionary operators: selection, crossover and mutation, the time complexity of them are O(K), O(KR), and O(KR), respectively. For the evaluation operator, we calculate a fitness for each hypothesis in the population, its time complexity is O(KRN). Thus, the whole time complexity of LCWF is O(TKRN + TK + 2TKR). If we only take the highest order terms, the time complexity of LCWF is O(TKRN). In the final version, we will provide the detailed time complexity analysis of LCWF in a new section called time complexity analysis. Thanks again for your valuable comments.

---

### Official Review · Reviewer_G1UX · 2024-03-23

**Q2-1 Originality-Novelty:** 3
**Q2-2 Correctness-Technical Quality:** 4
**Q2-5 Clarity Of Writing:** 4

**Q1 Summary And Contributions:**

Motivation:
This paper aims to solve the problem of multiple noisy labels obtained from crowdsourcing workers impacting the process of label integration. Label integration is a method utilized to infer unknown true labels. Existing research predominantly focuses on developing more complex and efficient label integration methods for more accurate inferences from noisy labels. However, there needs to be more research on purifying noisy labels prior to label integration. This paper proposes an innovative worker filtering algorithm to purify noisy labels.

Key Contributions:
This paper introduces a novel approach that purifies noisy labels before label integration to enhance the effectiveness of label integration. This approach contrasts with the common practice of developing more complex label integration methods. Specifically, it presents the Label Consistency-based Worker Filtering (LCWF) algorithm for purifying noisy labels in existing label integration methods. This approach transforms the worker filtering issue into a discrete optimization problem where label consistency dictates the fitness function. They then employ a genetic algorithm to discover the optimal solution. They demonstrate the efficacy of LCWF through experiments on both simulated and real-world datasets. The results indicate that LCWF can effectively purify noisy labels and significantly improve the integration accuracy of current label integration methods.

**Q2-3 Extent To Which Claims Are Supported By Evidence:**

3: Good: the main claims are supported by convincing evidence (in the form of adequate experimental evaluation, proofs, (pseudo-)code, references, assumptions).

**Q2-4 Reproducibility:**

3: Good: key resources (e.g. proofs, code, data) are available and key details (e.g. proofs, experimental setup) are sufficiently well-described for competent researchers to confidently reproduce the main results.

**Q3 Main Strengths:**

1. The idea of LCWF offers an effective method for purifying noisy labels. It innovatively transforms the problem of worker filtering into a discrete optimization problem, providing a new view of enhancing the performance of label integration.
2. The paper conducts tests on simulated and real-world datasets, proving its effectiveness in purifying noisy labels and improving the accuracy of existing label integration methods.
3. LCWF method demonstrates a notable increase in the accuracy of integrated labels, which is a crucial metric in label integration tasks.
The paper provides practical solutions to solve real-world problems in the field of label integration.

**Q4 Main Weakness:**

1. The simulations might benefit from including a variety of worker quantity scenarios to test the LCWF algorithm’s adaptability.
2. Detailed case studies could help better understand the algorithm’s practical implications and limitations.

**Q5 Detailed Comments To The Authors:**

This article explores the Label Consistency-based Worker Filtering (LCWF) method, which contributes to the research of label integration methods by purifying noisy labels before the label integration process. The paper's novelty is impressive, with fluent expression and comprehensive experiments. However, the authors could further optimize in a few areas:
In Section 3.2, where the authors transform worker filtering into a discrete optimization problem using label consistency, additional explanations or examples could be provided for better reader comprehension.
In Section 4.5, the discussion section, the authors could delve deeper into the potential limitations of their method. Specifically, exploring how the content of labels might influence worker quality assessment could offer broader insights. This might include considering how different labeling tasks could affect the effectiveness of the LCWF algorithm, thereby enriching the paper's applicability in various crowdsourcing scenarios.

**Q9 Complying With Reviewing Instructions:**

Yes

---

> ### Author Rebuttal · Authors · 2024-04-06
>
> Reviewer G1UX
> Q1: The simulations might benefit from including a variety of worker quantity scenarios to test the LCWF algorithm’s adaptability.
>
> Author Response:
> Thanks for your valuable comments. Indeed, in the real-world crowdsourcing scenarios, the worker quantity scenarios are complex and diverse. Actually, we have conducted the experiments on different worker quantity from 5 to 15 to test the performance of LCWF. We can nearly draw the same conclusions that our proposed method can improve the integration accuracy of existing label integration methods. Due to the page limitation, we do not provide all the experimental results in this version of the paper and only exhibit the experimental results when worker quantity is 9, which is near to the general real-world scenarios. In the final version, we will explicitly introduce these experiments on different worker quantity to demonstrate the adaptability of LCWF. Thanks again for your valuable comments.
>
> Reviewer G1UX
> Q2: Detailed case studies could help better understand the algorithm’s practical implications and limitations.
>
> Author Response:
> Thanks for your valuable comments. Indeed, detailed case studies can help better express our methods. In section 3.1, we have provided an example to express the motivation of LCWF to transform worker filtering into a discrete optimization problem using label consistency. In this example, we assume that there are five instances and five different workers have annotated them. The original multiple noisy labels and the multiple noisy labels after filtering a low-quality worker and a high-quality worker are provided in Table 1, Table 2 and Table 3, respectively. From these Tables, we can see that if a low-quality worker is filtered, higher label consistency is obtained after worker filtering, otherwise lower label consistency is obtained. That’s why we can use the label consistency to evaluate the fitness of different solutions of worker filtering and transform the worker filtering problem into a discrete optimization problem to solve.
> In the final version of the paper, we will further explain the practical calculation of the fitness of different hypotheses based on this example to show the practical implications and limitations of LCWF. Thanks again for your valuable comments.
>
> Reviewer G1UX
> Q3: In Section 4.5, the discussion section, the authors could delve deeper into the potential limitations of their method. Specifically, exploring how the content of labels might influence worker quality assessment could offer broader insights. This might include considering how different labeling tasks could affect the effectiveness of the LCWF algorithm, thereby enriching the paper's applicability in various crowdsourcing scenarios.
>
> Author Response:
> Thanks for your valuable comments. In section 4.5, we compactly discuss two special crowdsourcing scenarios: most workers show low-quality and worker collusion. When most workers show low-quality, then most labels annotated on the same instances may be incorrect.  Thus, high-quality worker can hardly reach consensus with other low-quality workers. In this case, if a high-quality worker is filtered, the label consistency of the multiple noisy labels may even improve. Based on the current fitness function, the hypotheses which filter high-quality workers can have a less fitness than the hypotheses which filter low-quality workers, which will lead to poor performance on filtering low-quality workers. As for worker collusion, it means that some workers provide totally consistent labels on most or even all of instances they have annotated. If the consistent labels of these collusive workers are correct, then worker collusion can be beneficial for LCWF to filtering other low-quality workers. However, if the consistent labels of these collusive workers are incorrect, then LCWF may mistake these workers as high-quality workers due to the high label consistency of their labels and filter the actual high-quality workers instead. Thus, the performance of LCWF is not stable when there is worker collusion.
> In the final version, we will add these discussion to section 4.5 to enrich the paper's applicability in various crowdsourcing scenarios. Thanks again for your valuable comments.

---

### Meta-Review · Area_Chair_Tfc3 · 2024-04-15

This paper targets learning with noisy labels in crowdsourcing scenarios. The authors propose purifying the labels before integrating them.

In reviews, most reviewers acknowledge the novel idea, the effectiveness of the method, and the quality of the writing. All reviewers rate positively on this paper.